# Evaluating Desk-Assisted Standing Techniques for Simulated Pregnant Conditions: An Experimental Study Using a Maternity-Simulation Jacket

**DOI:** 10.3390/healthcare12090931

**Published:** 2024-05-01

**Authors:** Kohei Uno, Kako Tsukioka, Hibiki Sakata, Tomoe Inoue-Hirakawa, Yusuke Matsui

**Affiliations:** 1Biomedical and Health Informatics Unit, Graduate School of Medicine, Nagoya University, 1-1-20 Daiko-Minami, Higashi-ku, Nagoya City 461-8673, Aichi, Japan; 2School of Health Sciences, Nagoya University, 1-1-20 Daiko-Minami, Higashi-ku, Nagoya City 461-8673, Aichi, Japan; 3Department of Integrated Health Sciences, Graduate School of Medicine, Nagoya University, 1-1-20 Daiko-Minami, Higashi-ku, Nagoya City 461-8673, Aichi, Japan; 4Institute for Glyco-Core Research, Tokai National Higher Education and Research System, Nagoya University, 1-7 Furo-cho, Chikusa-ku, Nagoya City 464-0814, Aichi, Japan

**Keywords:** sit-to-stand, pregnancy, maternity-simulation jacket, center of pressure, Azure Kinect, surface electromyography

## Abstract

Lower back pain, a common issue among pregnant women, often complicates daily activities like standing up from a chair. Therefore, research into the standing motion of pregnant women is important, and many research studies have already been conducted. However, many of these studies were conducted in highly controlled environments, overlooking everyday scenarios such as using a desk for support when standing up, and their effects have not been adequately tested. To address this gap, we measured multimodal signals for a sit-to-stand (STS) movement with hand assistance and verified the changes using a *t*-test. To avoid imposing strain on pregnant women, we used 10 non-diseased young adults who wore jackets designed to simulate pregnancy conditions, thus allowing for more comprehensive and rigorous experimentation. We attached surface electromyography (sEMG) sensors to the erector spinae muscles of participants and measured changes in muscle activity, skeletal positioning, and center of pressure both before and after wearing a Maternity-Simulation Jacket. Our analysis showed that the jacket successfully mimicked key aspects of the movement patterns typical in pregnant women. These results highlight the possibility of developing practical strategies that more accurately mirror the real-life scenarios met by pregnant women, enriching the current research on their STS movement.

## 1. Introduction

Pregnancy induces significant hormonal changes [1] and a shift in the center of gravity [2] due to uterine development, directly contributing to low back pain (LBP) experienced by many pregnant women [3]. Therefore, it is an important challenge to clarify the causes of LBP in pregnant women and to prevent it. Lifting heavy objects is generally considered a risk factor for pregnancy-related LBP [4,5,6]. However, cohort studies have revealed that a considerable proportion of pregnant women suffer from LBP in various daily activities, including sitting, standing up from a chair, and turning over while lying down [7]. Therefore, these daily activities can be the cause of their LBP. In other words, it is necessary to consider not only high-intensity activities but also the influence of basic activities of daily living (BADL) on pregnancy-related LBP. In pregnancy-related LBP research, sit-to-stand (STS) movements among BADL are important because they are related to LBP [8]. STS is a task of standing up from a chair, and by conducting experiments on elderly people, patients, and pregnant women, it is possible to understand the behavioral characteristics.

In this study, to assess the load on the lumbar during the STS movement, we measure the activity of the spinal erector muscles using surface electromyography (sEMG) [9]. A direct causal relationship between the erector spinae muscles and LBP has not been proven. However, many studies about the exoskeleton have determined that the load of lumber is reduced if the activity of the erector spinae muscles is suppressed [10,11,12,13].

Previous research has extensively covered the relationship between STS and pregnant women [14,15,16]. However, in these studies, subjects stood without assistance, that is without holding on to tables or handrails, a significant oversight considering the realities of pregnant women’s living condition. To address this gap, our study specifically investigates the biomechanics of standing up using a table, which is commonplace in home environments such as living rooms. We aim to investigate how different methods of hand support, specifically placing hands on a table versus placing hands on the knees, affect the load on the lumbar region during the process of STS.

Experiments on pregnant women also present other difficulties. This means that negative effects on pregnant women and their fetuses must be avoided, and pregnant women cannot be asked to perform heavy tasks. To address this issue, we adopted a new approach by having non-pregnant women wear a Maternity-Simulation Jacket to simulate the state of pregnancy. This method could help in safely assessing the load without subjecting pregnant women to potential risks. However, even if the jacket is used, there is no guarantee that the characteristics of pregnant women during STS will be truly reproduced.

There are two primary purposes for this study. The first is to verify whether the muscle load during STS movements varies depending on the way the subjects place their hands on to the table when standing up. The second purpose is to confirm whether the characteristics of a pregnant woman can be reproduced by a Maternity-Simulation Jacket. Understanding these factors is crucial as they can inform more effective guidelines and care strategies for pregnant women during standing up, thus enhancing their overall well-being during this critical period. This study also has the potential to expand on the experimental studies on pregnant women.

## 2. Materials and Methods

### 2.1. Participants

We recruited 10 non-diseased young women from a university community who consented to participating in the study (Table 1). The sample size was determined with reference to sample sizes in past STS studies [16,17,18,19].

Among the 10 participants, 4 had a habit of exercising, while 6 did not. The participants were orally and in writing informed about the purpose, methods, risks, concerns, and anonymity of the experiment, and written consent was obtained. The inclusion criteria were females affiliated with Nagoya University who provided consent to participate in this study. The exclusion criteria were females under 18 years of age, or with a history of lumbar conditions, or currently experiencing any form of bodily pain. The Ethical Research Committee for Health Sciences and Clinical Research of Nagoya University approved this research (protocol code: 23-702).

### 2.2. Experimental Environment and Data Acquisition

The center of foot pressure (COP) was estimated using the Wii Balance Board (Figure 1a). The sEMG sensors (BioSignalsPlux) were attached 4 cm from the third and fourth lumbar vertebrae to measure the muscle activities during standing up (Figure 1b). Joint coordinates were estimated using Azure Kinect and Azure Kinect Body Tracking [20,21,22]. In addition, the subjects’ age, height, weight, and exercise habits were obtained through a questionnaire.

Azure Kinect was set up outside an approximately 3 m square area. In front of the Kinect, a chair with a seat height of about 40 cm and a desk with a tabletop height of about 70 cm were placed. The Wii Balance Board was positioned on the floor in front of the chair (Figure 1d,e).

### 2.3. Experimental Procedure

After explaining the purpose of the study to the subjects using an instruction manual and obtaining their consent to participate, the subjects filled out a questionnaire with their age, height, weight, and exercise habits. Then, sEMG sensors were attached to their lumbar region. The subjects wore a non-bulky top and trousers and stood barefoot on the Wii Balance Board, positioning the insides of their feet 10–15 cm apart, perpendicular to Azure Kinect. STS movements were performed in two patterns. Both patterns consist of three conditions: both-hands-on-knees, one-hand-on-table, and both-hands-on-table. The difference is whether a Maternity-Simulation Jacket is worn. We randomly decided which pattern to do first. All conditions start from a standing position, sit down, pause for about 1 s, and then stand up.

Repeat 3 times to stand up with your hands on your knees without the Maternity-Simulation Jacket (both-hands-on-knees). Then, repeat 3 times to stand up, place your right hand on the right side of the desk and stand up (one-hand-on-table). Then, repeat 3 times to stand up and place your hands on the front desk (both-hands-on-table).Repeat 3 times to stand up with your hands on your knees with the Maternity-Simulation Jacket (both-hands-on-knees). Then, repeat 3 times to stand up, place your right hand on the right side of the desk and stand up (one-hand-on-table). Then, repeat 3 times to stand up and place your hands on the front desk (both-hands-on-table).

For both-hands-on-knees, subjects were instructed to grasp their knee joints with both hands. For one-hand-on-table, they were instructed to place their right hand on the right side of the desk without a specific hand position and to let their left hand hang naturally. For both-hands-on-table, they were instructed to place both hands shoulder-width apart on the desk in front. In all conditions, subjects were instructed to rise by applying weight to their hands and to straighten their back during the action. A Maternity- Simulation Jacket, simulating the weight of a late-term pregnancy (8–9 months), weighing approximately 7.2 kg, was used. The abdominal circumference was measured before and after wearing the jacket, ensuring a difference of about 33 cm.

To synchronize data, the following procedures were conducted before the six STS actions: For synchronizing the Azure Kinect data and the sEMG sensors, the subject raised their left hand and pressed a button on Biosignalsplux three times. To synchronize the Azure Kinect and the Wii Balance Board, they performed toe stands three times.

### 2.4. Statistical Analysis

All statistical analyses were performed using the programming software R version 4.2.3. Statistical analysis was conducted according to the flowchart shown in Figure 2.

We conducted an integrated analysis of the EMG measured by sEMG, the COP measured by a Wii Balance Board, and the joint coordinates estimated by Azure Kinect. Initially, we preprocessed the EMG signals.

#### 2.4.1. Rectification

The EMGs of the left and right erector spinae muscles were measured at T time points. Let the raw EMG signal vector be $x_{raw}\in R^{T}$. We converted $x_{raw}$ to the vector $x_{mean}$ with mean 0 and obtained the full-wave rectification of the EMG signal as below:$$x_{rect}=\left|x_{raw}-mean\left(x_{raw}\right)\right|\;.$$

#### 2.4.2. Filtering

In EMG analysis, it is common to analyze only frequencies of interest. Thus, we obtained the signals $x_{filt}$ with a limited frequency band using a bandpass filter (5–30 Hz).

#### 2.4.3. Moving Average

We smoothed the filtered signal by moving average. When time point t of the filtered signal $x_{filt}$ is expressed as $x_{filt}(t)$, we obtained the moving averaged vector as below:$$x_{mv}\left(t\right)=\frac{1}{2L+1}\sum_{i=-L}^{L}x_{filt}\left(t-i\right)\;,$$
where L determines the size of the sliding window. In this study, we set L to 10.

#### 2.4.4. Normalization

Normally, EMG signals are scaled so that the maximum value is 100 and the minimum value is 0. Therefore, we scaled the signal by normalizing $x_{mv}$. If the Maximum Voluntary Contraction (MVC) could be measured, it is common to set MVC to 100(%). However, MVC could not be measured in this measurement. Thus, we set the maximum value of $x_{mv}$ to 100 and the minimum value of $x_{mv}$ to 0,
$$x_{scale}=\frac{x_{mv}-\min\left(x_{mv}\right)}{\max\left(x_{mv}-\min\left(x_{mv}\right)\right)}\times 100\;.$$

#### 2.4.5. Resampling

In this study, the EMG signals were measured at 1000 Hz, while the COP signal was measured at 100 Hz, and the joint coordinate signals were measured at 30 Hz. For integrated analysis, all signals should be the same time interval. Therefore, we aligned these signals to 1000 Hz. First, we considered the signals as functions. Next, we resampled them as 1000 Hz signals by reassigning the time points every 1 ms.

#### 2.4.6. Signal Synchronization

It is necessary to synchronize the EMG, COP, and joint coordinate signals because the signals have different measurement start points. Therefore, we achieved signal synchronization by capturing the characteristic movements of the EMG, COP, and joint coordinate signals. Specifically, three types of signals were synchronized by combining EMG and joint coordinate synchronization, and COP and joint coordinate synchronization.

As mentioned in Section 2.3, the subjects repeated the process three times by raising their left hand and pressing a button switch on the Biosignalsplux connected to the EMG. Therefore, let $D\left(cj_{1},\;EMG_{1}\right)$ be the difference between the time when the left hand’s z-axis coordinate reaches its maximum value for the first time and the time when the button switch is pressed for the first time. The average value of these 3 times was defined as the difference between the joint coordinates and EMG,
$$D\left(cj,\;EMG\right)=\frac{1}{3}\sum_{k=1}^{3}D\left(cj_{k},\;EMG_{k}\right)\;.$$

The subjects then stood on their tiptoes three times. Therefore, let $D\left(cj_{1},\;COP_{1}\right)$ be the difference between the point at which the y-axis coordinate of the head reaches its maximum value for the first time and the point at which the y-axis coordinate of the COP, in other words, the point at which the forward coordinate reaches its maximum value for the first time. The average value of these three times was defined as the difference between joint coordinates and the COP,
$$D\left(cj,COP\right)=\frac{1}{3}\sum_{k=1}^{3}D\left(cj_{k},\;COP_{k}\right)\;.$$

#### 2.4.7. Segmentation

We segmented the standing-up phase which involved identifying the time point of the minimal y-coordinate of the pelvis as the sitting state and the time point of the maximal y-coordinate of the head as the standing state.

#### 2.4.8. Exclude Outlier

The coordinates of all coordinates estimated by Azure Kinect may be missing. If all joint coordinates are missing values half of the time in one epoch, exclude that epoch.

#### 2.4.9. Comparison via *t*-Test

We compared the jacket-worn and non-worn conditions using *t*-tests for each value.

## 3. Results

### 3.1. Understanding the Data through Visualization

For data analysis in this study, it is important to understand the outline of the EMG, COP, and joint coordinate. Therefore, we grasped trends by visualizing each set of data. First, the time points of the segmented data were normalized to 100. Then, to enable visualization using the same scale, the right erector spinae EMG, the weight measured by the foot pressure meter, and the head height (y-axis coordinate) measured by Azure Kinect were z-transformed. Three conditions (both-hands-on-knees, one-hand-on-table, both-hands-on-table) for one subject were plotted (Figure 3).

Comparing the EMGs, in the case of both-hands-on-knees and both-hands-on-table, the EMGs have two peaks, one at the beginning of the rise and the other at the end. This is because when using both hands, the subject initially leans forward and then stands up. On the other hand, in the case of one-hand-on-table, there is only one EMG peak. This is thought to be due to standing up without leaning forward significantly. The results are consistent with those of previous studies without hand assistance [24,25].

### 3.2. Comparison of Muscle Load during STS Using EMG Analysis

In this study, erector spinae muscle loads during STS were evaluated from the sEMG sensors attached to the left and right sides of the lumbar region. The evaluation used the iEMG (integral electromyogram), which is calculated by estimating each interval when standing up and integrating the EMG within that interval, and the maximum value of the EMG.

When we examined whether there was a difference between the three conditions when wearing the Maternity-Simulation Jacket by analysis of variance, there was no significant difference in either the iEMG or maximum EMG. Therefore, it cannot be said that the load varies greatly depending on the type of assistance.

When comparing whether there was a difference between the left and right iEMG using a paired *t*-test, the right iEMG was significantly larger in the one-hand-on-table condition (*p* = 0.048) (Figure 4a). Similarly, when the left–right difference in the maximum EMG was compared using the paired *t*-test, the left maximum EMG was significantly larger in the both-hands-on-table condition (*p* = 0.044) (Figure 4b). Thus, there was a difference in the iEMG between the left erector spinae and right erector spinae under each condition, and we found that the right iEMG was larger under the one-hand-on-table condition. Such asymmetric loading has also been confirmed in similar studies. When pregnant women in the third trimester of pregnancy were tested on standing up using a handrail, it was found that assisting with one hand placed an asymmetrical load on the legs [26,27]. Such an asymmetrical load may lead to lower back pain [28].

### 3.3. Comparison of Trunk Flexion Angle during STS

Next, we compared the maximum flexion angle of the trunk during STS under three conditions using a *t*-test while wearing the Maternity-Simulation Jacket. The bending angle of the trunk was defined as the angle between the vector obtained by connecting the coordinates of the pelvis and neck estimated by Azure Kinect Body Tracking and the vector directed from the pelvis to the ceiling (Figure 5a). Since this was a three-condition test, a corrected p-value was calculated using the Benjamini–Hochberg method. The results showed that in the one-hand-on-table condition, when standing up with the right hand on the desk on the right side, the bending angle was significantly smaller than in the other conditions. It has been reported that the bending angle is related to the load on the lumbar [29,30], so it is possible that the “one-hand-on-table” condition, which has a small bending angle, places less stress on the lumbar region.

### 3.4. Reproducing the Pregnant State

The results so far compare the lumbar region load and trunk angle when wearing the Maternity-Simulation Jacket. However, it is unknown whether the jacket can imitate the condition of a pregnant woman. One of the purposes of this study is to verify whether the characteristics of STS in pregnant women can be reproduced using the jacket. Therefore, we will examine whether the jacket can reproduce the characteristics of pregnant women that have been revealed in past research.

First, we compared the EMG of erector spinae. As a result of the *t*-test, the iEMG was significantly larger when wearing the jacket under all conditions (Figure 6a). The maximum EMG was significantly larger when wearing the jacket, except for both-hands-on-table (Figure 6b). Thus, the iEMGs when wearing the jacket were significantly larger than the iEMGs when not wearing the jacket under all conditions. These results suggest that the loads on the lumbar region increased when wearing the jacket. This is one of the characteristics seen in pregnant women, and this characteristic could be imitated [31].

Next, the average COP in the forward and backward directions during standing up was compared using a *t*-test. As a result, under all conditions, the COP was significantly biased toward the rear when wearing the jacket (Figure 6c). This is thought to be because the COP moves backward as a result of adaptation to the forward movement of the center of gravity when wearing the jacket. This is a characteristic that has also been observed in late pregnancy and can be mimicked by wearing the jacket [32].

Finally, the maximum flexion angle of the trunk during standing up was compared using a *t*-test. As a result, it could not be said that there was a significant difference in all conditions depending on whether a jacket was worn or not (Figure 6d).

## 4. Discussion

In this study, we investigated the causes of LBP by measuring the COP, erector spinae muscle loads, and joint coordinates of non-diseased young women when they stood up from their chairs with and without a Maternity-Simulation Jacket. The jacket successfully reproduced some of the biomechanical changes experienced by pregnant women. We also observed distinct patterns in the EMG, COP, and joint coordinates under different conditions. Notably, standing up with one-hand-on-desk imposed an asymmetrical load on the lumbar region. Furthermore, we found that the maximum trunk flexion angle was significantly smaller in the one-hand-on-table condition compared to other conditions.

One of the purposes of our study is to verify whether a Maternity-Simulation Jacket can reproduce the characteristics of pregnant women. In Section 3.4, we demonstrate how the jacket mimics the characteristics. We examined three characteristics of pregnant women: the EMG, COP, and maximum trunk flexion angle. We verified the reproducibility of the EMG from two perspectives, the iEMG and the maximum EMG, and found that EMG activity increased relatively in all conditions with one exception. Similarly, the COP is located at the rear in all conditions. However, the maximum trunk flexion angle could not be reproduced. It is known that the trunk angle of pregnant women usually decreases during STS compared to non-pregnant women [33]. This phenomenon is not limited to STS; it is also known that the ROM of the trunk angle during walking decreases [34]. This has been pointed out to be an adaptive behavior that occurs due to loss of balance stability due to increased abdominal weight [35,36]. However, the jacket clearly moved the center of gravity forward, and the balance stability must have been lost. Psychological factors may be the reason why the trunk angle does not change despite this. Some people may consider that flexing the trunk puts pressure on the abdomen, which may have a negative impact on the fetus [37]. However, wearing a jacket cannot imitate such psychological aspects, so it is possible that the trunk angle could not be reproduced. Alternatively, the maximum flexion angle of the trunk may not have been able to change because the subject was instructed to place his hands on the desk and the position of his feet was fixed. It is known that characteristics change depending on the state before standing up, such as the height of the chair, so the results may change if the subject is allowed to freely choose how to stand up [15,38,39]

The other purpose of this study was to verify whether placing subjects’ hands on a desk when standing up can reduce the load on the lower back. We compared EMG activity under three conditions: both-hands-on-knees, one-hand-on-table, and both-hands-on-table, but there were no significant differences. On the one hand, one-hand-on-table is a condition in which the maximum trunk flexion angle is significantly small, but on the other hand, the EMG activity is asymmetrical. This conflicting result may require more detailed analysis. In other words, it is pointed out that analysis based only on electromyography is insufficient, and it is necessary to calculate moments that also combine information such as lumbar flexion [40]. Therefore, more detailed analysis will be required in the future.

The limitations of this study include the small sample size of the subjects and limited characteristics such as age and BMI. This study consists of 10 subjects. There are 30 epochs because each condition was started three times, but this is not a sufficient sample size. Therefore, in order to more precisely verify the findings obtained in this study, it is essential to construct an experimental design through a prior power analysis. For example, the results of this study are strongly influenced by body size [41]. Depending on the subjects’ body size, the results obtained in this study may not be replicated. The BMI of the subjects was 17.3 to 23.1, and there were no obese subjects, so it is thought that it is necessary to obtain data that include obese people. Increasing physical strength through exercise may prevent LBP and pain-related disorders in pregnant women, so it will be necessary to compare women with and without exercise habits in the future. Moreover, in this study, we recruited a homogeneous sample because the results may vary depending on body size and exercise experience. However, this limits the results of this study. The average age of mothers at the birth of their first child in 2021 in Japan is 30.9 years old [42], but the subjects in this study are between 20 and 22 years old, which is different from the actual age of pregnant women. Therefore, it is considered necessary to select subjects who are around 30 years old.

## 5. Conclusions

In conclusion, our study was able to safely replicate the conditions experienced by pregnant women using a Maternity-Simulation Jacket. These results underscore the potential for developing practical, real-world strategies to alleviate the physical strain associated with STS movements in pregnant women.

## Figures and Tables

**Figure 1 healthcare-12-00931-f001:**
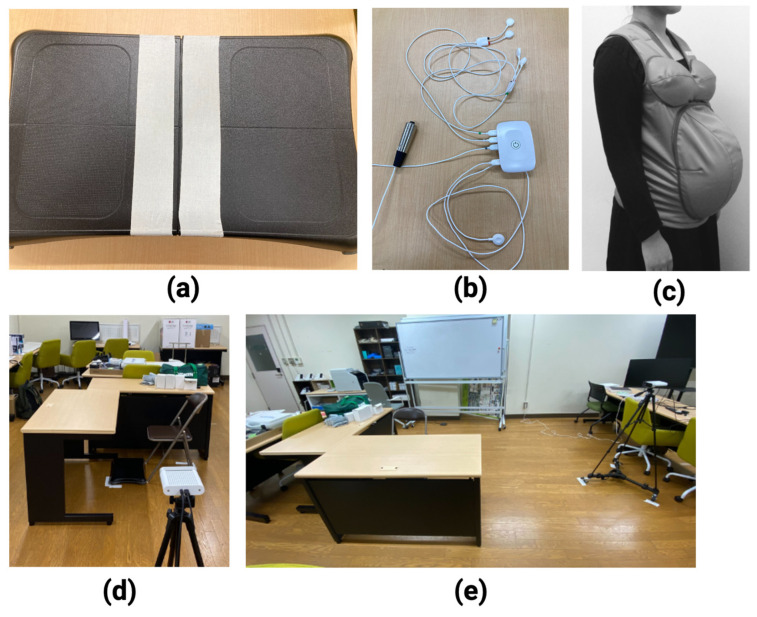
Experimental environment: (**a**) Wii Balance Board to measure the center of pressure; (**b**) Biosignalsplux connected with sEMG sensors and switch for signal synchronization to measure erector spinae muscles activity; (**c**) Maternity-Simulation Jacket [23]; (**d**) Azure Kinect to measure from the sagittal plane; (**e**) desks were set up in front of and to the right of the chair.

**Figure 2 healthcare-12-00931-f002:**
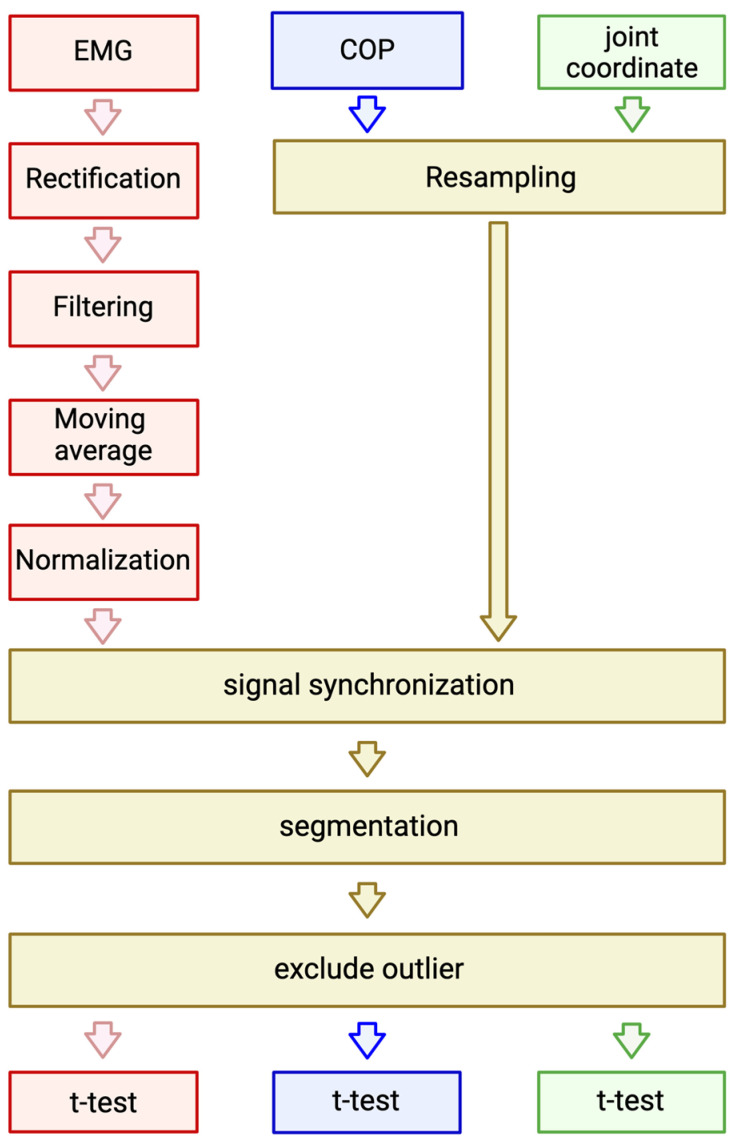
Data analysis flowchart.

**Figure 3 healthcare-12-00931-f003:**
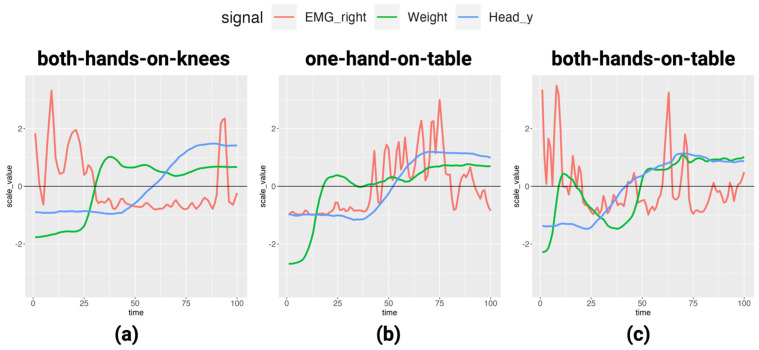
Relative time-series changes during STS in three signals, right lumbar EMG, weight measured with Wii Balance Board, and head y-axis coordinate: (**a**) time-series changes under both-hands-on-knees condition, (**b**) time-series changes under one-hand-on-table condition, (**c**) time-series changes under both-hands-on-table condition.

**Figure 4 healthcare-12-00931-f004:**
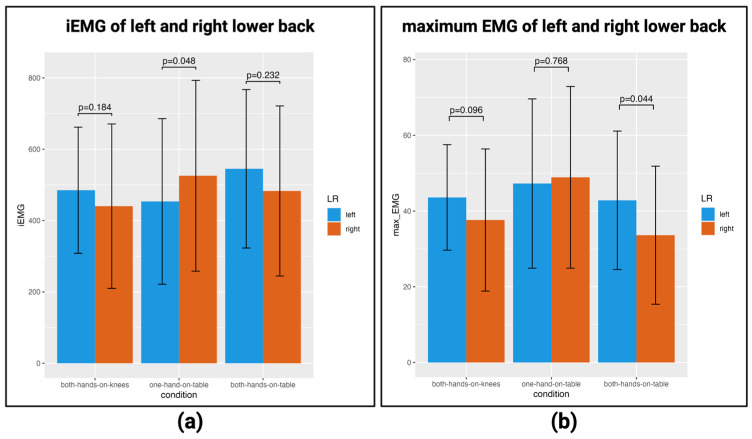
Comparison of left and right EMG: (**a**) left and right iEMG for each condition; (**b**) left and right maximum EMG for each condition. Appendix A show the details.

**Figure 5 healthcare-12-00931-f005:**
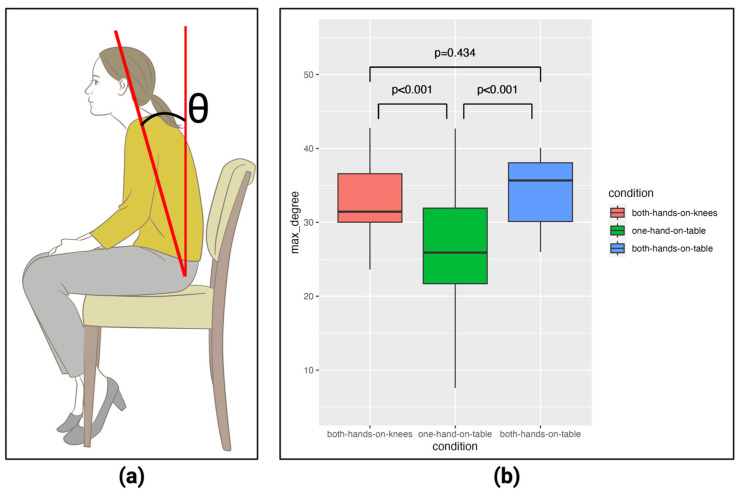
(**a**) The way to calculate the trunk angle and (**b**) the comparison of maximum flexion angle between conditions. Appendix A shows the details.

**Figure 6 healthcare-12-00931-f006:**
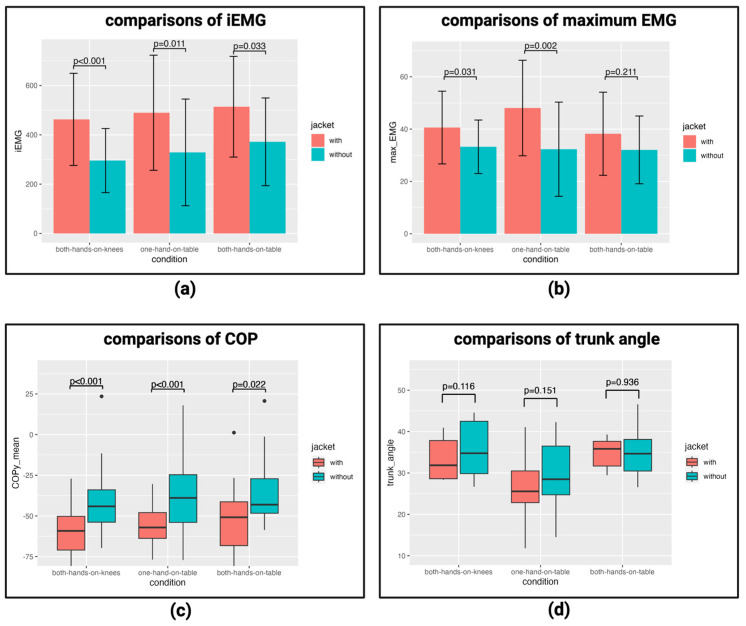
Comparison between wearing the jacket and not wearing the jacket: (**a**) comparison of iEMG; (**b**) comparison of maximum EMG; (**c**) comparison of average COP; (**d**) comparison of maximum trunk flexion angle. Appendix A provide the details.

**Table 1 healthcare-12-00931-t001:** Characteristics (mean and standard deviation) of the study sample (N=10).

Variables	Mean	SD
Age (year)	21.10	0.57
Height (cm)	157.82	3.60
Weight (kg)	48.66	4.78
$\mathrm{BMI}\;(\mathrm{kg}/m^{2}$)	19.53	1.75

## Data Availability

The data are available upon reasonable request.

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
