# Peer review of "Evaluating Desk-Assisted Standing Techniques for Simulated Pregnant Conditions: An Experimental Study Using a Maternity-Simulation Jacket"

_healthcare, 2024, doi:10.3390/healthcare12090931_

Round 1
Reviewer 1 Report
Comments and Suggestions for Authors
INTRODUCTION
1. I believe that the relationship between pain and stress, anxiety and/or depression has nothing to do with what we keep on talking about afterwards. I recommend rewriting this paragraph
2. You talk about general activity but then you only evaluate sitting and standing up?
3. Why is it assumed that more activity in the spinal erectors is related to more low back pain?
4. Relationship with muscle torque or muscle strength?
5. It is coherent that there is an increase in muscle tone due to the advance of the center of gravity, hormonal alteration and uterine development, up to this point I agree, however for the analysis of pain you only take into account the electromyographic activity of the lumbar musculature? And only how it is activated in sitting and standing up more specifically?
6. If previous research already fulfills this relationship, what does this article contribute to current science?
7. It is not very clear what the authors mean by the subjects remain unaided.
8. The introduction has nothing to do with the objective stated at the end of it. I recommend rewriting it in its entirety.
9. In the case that the woman before pregnancy trains and has a greater activity that is in charge of providing lumbar support, don't you think that she will also be electromyographically more active and it does not have to be causally correlated with pain?
10. I do not understand why the introduction of your study includes results. Each section determines the information that should be included
METHODS
Why only 10 participants? Has a previous statistical analysis been made or has a previous article been used as a reference?
2. Is the age of the persons included in the study representative of the gestational age? Should this be taken into account for reproducibility and external validity?
3. Why only subjects without pain are taken? Would it then only be possible to apply these results to pregnant women who did not have low back pain prior to pregnancy?
4. Why do you not include photos of the simulation suit used but of the rest of the material? It would be of interest
RESULTS
1. If there is no change between using the suit or not electromyographically speaking, what does this study contribute?
2. It would be of interest if the data were included in tables, even if only as supplementary material to help readers understand more easily.
3. What does the maximum trunk flexion angle contribute? So far nothing has been mentioned about this
DISCUSSION
1. More studies should be included to discuss and therefore more bibliography should be included.
Author Response
Thank you for your insightful comments and suggestions.
We attached the file. Please read it.

Reviewer 2 Report
Comments and Suggestions for Authors
Dear colleagues!
Thanks for the interesting and relevant research.
The strengths of the work are related to Lower back pain, which worries pregnant women. The particular value of the work is obvious in the competent construction of the model and its dynamic load, which made it possible to perform the experiment under conditions close to natural ones.
By way of discussion, I would like to ask a few questions.
1. You only have 10 participants - how did you determine the sample size?
2. How did you synchronize the actions of the participants? or did you have a certain load interval within which they operated?
3. What clinical recommendations can you give regarding the norm under study?
4. Why were there no people with high body mass index in the study?
Author Response
Dear Reviewer,
Thank you very much for reviewing our manuscript and providing valuable and constructive comments.
We have responded to all of the points raised by you and the other reviewers, and revised the manuscript accordingly.

Reviewer 3 Report
Comments and Suggestions for Authors
Dear authors;
First of all, I would like to thank the authors for their overall efforts during the study. It is a good and clearly described study. The topic is interesting, yet, in its current form, this paper cannot be considered for publication. However, I see value in the research approach and strongly encourage the authors to address the following points.
Abstract
· Line 22: Please provide the number of sample size
· Line 21: Try to explain methods in detail.
Introduction
· The introduction is not good enough to understand the topic.
· Line 49: Please revise this statement. “We focus….”
· Line 50: This sentence also does not match with the typical pattern of the Introduction. Try to use passive sentences.
· Line 70: Why did the authors provide the study results at the end of the Introduction? Please remove them and add study hypotheses.
Materials and Methods
· Line 78: Did you perform a priori power analyses to estimate the number of samples? Why 10 participants?
· Please provide inclusion/exculuısion criteria clearly.
· Add an ethical statement just after the statement of “written consent”.
· Please revise the statistical analysis section and try to be more clear.
Results
· I am fine with the results, tables, and figures. They reflect the measured variables and related findings.
Discussion & Conclusions
· The discussion is well-structured and written.
· Please revise the statement related to Section 3.1. and the rest of them throughout the Discussion. I suggest removing such statements.
· The authors should provide more citations in the Discussion. I can not see a real “discussion” here. This is the main problem of the paper. Please better discuss the study findings.
· The conclusions are too long for the readers. So, please revise it and pay attention to the main and crucial findings of the study.
· The references are not really enough in terms of both quality and quantity for such research.
Comments on the Quality of English Language
Moderate editing of English language required
Author Response
We would like to thank you for your thoughtful and constructive comments. All suggestions were considered and incorporated into the revised manuscript. Your review has significantly improved our manuscript. We have attached the file. We would appreciate it if you could read it.
